# Nitrogen Transport/Deposition from Paddy Ecosystem and Potential Pollution Risk Period in Southwest China

**Shufang Guo [1,2,†], Tiezhu Yan [1,†], Limei Zhai [1,*], Haw Yen [3], Jian Liu [4], Wenchao Li [1] and Hongbin Liu [1]**

[1] Key Laboratory of Nonpoint Source Pollution Control, Ministry of Agriculture and Rural Affairs/Institute of Agricultural Resources and Regional Planning, Chinese Academy of Agricultural Sciences, Beijing 100081, China; guosf12b@163.com (S.G.); yantiezhu@caas.cn (T.Y.); dachao279@126.com (W.L.); liuhognbin@caas.cn (H.L.)

[2] Institute of Agricultural Environment and Resources, Yunnan Academy of Agricultural Sciences, Kunming 650201, China

[3] Blackland Research and Extension Center, Texas A&M Agrilife Research, Texas A&M University, Temple, TX 76502, USA; haw.yen@gmail.com

[4] School of Environment and Sustainability, Global Institute for Water Security, University of Saskatchewan, Saskatoon, SK S7N 0X4, Canada; jianliu1985yy@163.com

* Correspondence: zhailimei@caas.cn

† These authors contributed equally to this work.

**Abstract:** Nitrogen (N) losses through runoff from cropland and atmospheric deposition contributed by agricultural $NH_3$ volatilization are important contributors to lake eutrophication and receive wide attention. Studies on the N runoff and atmospheric N deposition from the paddy ecosystem and how the agriculture-derived N deposition was related to $NH_3$ volatilization were conducted in the paddy ecosystem in the Erhai Lake Watershed in southwest China. The critical period (CP) with a relatively high total N (TN) and $NH_4^+$-N deposition occurred in the fertilization period and continued one week after the completion of fertilizer application, and the CP period for N loss through surface runoff was one week longer than that for deposition. Especially, the mean depositions of $NH_4^+$-N in the CP period were substantially higher than those in the subsequent period ($p < 0.01$). Moreover, agriculture-derived $NH_4^+$ contributed more than 54% of the total $NH_4^+$-N deposition in the CP period, being positively related to $NH_3$ volatilization from cropland soil ($p < 0.05$). The N concentrations were higher in the outlet water of ditches and runoff in May than in other months due to fertilization and irrigation. Therefore, to reduce the agricultural N losses and improve lake water quality, it is important to both reduce agricultural $NH_4^+$-N deposition from $NH_3$ volatilization and intercept water flow from the paddy fields into drainage ditches during the CP.

**Keywords:** nonpoint source pollution; critical period; nitrogen deposition; $NH_3$ emissions; surface runoff; paddy field

## 1. Introduction

Eutrophication of lakes and reservoirs resulting from an excess input of nitrogen (N) was recognized as an important water environmental problem worldwide [1–4]. Especially, the eutrophication of inland freshwater lakes threatens the ecological functions of lakes, freshwater supply and flood mitigation [5]. A total of 54%, 53%, 46% and 28% of lakes in Asia, Europe, North America and Africa face eutrophication problems, respectively, according to statistics from the Water Research Commission, South Africa [6]. In China, approximately one third of all lakes are freshwater ones, but currently, most freshwater lakes are in mesotrophic or eutrophic condition [7]. A large N load delivered to watersheds was one of the primary causes for the frequent occurrence of algae blooming in lakes [5]. Agriculture was estimated to be the source of 57% of the N entering water bodies in China and 37%–82% of N losses into surface waters in western Europe [8,9].

Therefore, agriculture activities and the associated N losses impose serious potential "downstream" pressure on aquatic environments [10,11].

Nitrogen outputs from cropland vary because of agricultural activities (e.g., seasonal fertilization) [12–15]. Previous studies showed that N losses in rice systems mainly occurred via leaching, gaseous losses and surface water runoff [16,17]. NH₃ is the primary form of reactive N from cropland [18,19]. NH₃ volatilization from paddy fields accounted for 17.7% (14.4–21.0%) of the N applied, primarily found within two weeks following each fertilization event [20]. For example, NH₃ concentrations increased sharply and reached peak values after N fertilizer application in both the North China Plain and in the double rice region of subtropical China [21,22]. Additionally, NH₃ was a critical contributor to N deposition in high NH₃ emission areas through dry and wet deposition [23]. Large NH₃ emission sources contributed to high N loads in nearby ecosystems through dry deposition [24,25]. The NH₄⁺-N and TN in wet deposition in a rice agroecosystem were both significantly positively related to NH₃ volatilization from paddy fields primarily because of fertilization [26]. Moreover, the use of N fertilizer was positively related to wet N deposition in China [27,28]. While studies of ecological effects of increased N deposition on the eutrophication of lakes could be found in literature [29,30], there were few studies that distinguish the N deposition derived from cropland (Refs.) [31,32]. Previous studies showed that the contribution of atmospheric N deposition reached approximately 16% of N input in the Taihu (eastern China) and Dianchi lakes (southwest China) [33,34]. However, those N depositions were influenced by multiple sources. Therefore, a quantitative evaluation of N deposition derived from cropland and a determination of the critical period of N deposition are useful to develop efficient strategies for alleviating the current condition of eutrophication in lakes.

Agricultural runoff was recognized as an important driver of water quality degradation in southwest China [35]. Lakes in southwest China, such as the Yunnan Plateau lakes, were in a typical initial stage of eutrophication, and they had also received more attention in socioeconomic development [36–39]. More than 78% of cropland in the Yunnan Plateau area was rice paddy fields [36]. Tang et al. [40] reported that the N load from farmland is one of the primary influencing factors of eutrophication, primarily from paddy land in the northern watershed of Erhai Lake. N losses via hydrological processes from agricultural land increased with an increase in N fertilizer application rates [41]. Moreover, N concentration in runoff increased considerably when runoff occurred immediately after fertilization [42,43] and thus had direct and great effects on water pollution [15]. However, there was no quantitative evaluation on the duration period of high N concentration in runoff in regional levels in the literature.

The rice growing season is from May to October (which is also the wet season), whereas precipitation and runoff were the two primary drivers for the N transfer to lakes. The primary goal of this study was to test the hypothesis that the contribution of cropland to N losses by runoff and deposition were different in time. Specifically, three objectives were defined: (1) understand the temporal change of N losses in runoff and N deposition after agricultural activities; (2) identify the high load period of N output via runoff and N deposition in an agricultural region; (3) explore the relationship between NH₃ emissions and N deposition in agricultural ecosystems and the contribution of farmland to N deposition.

## 2. Materials and Methods

### 2.1. Site Description

Erhai Lake is the second largest plateau freshwater lake in a typical agriculture-dominated watershed in the Yunnan Plateau area, which is in the Mesotrophic status [1]. The percentage of agricultural areas, water bodies (river/lakes) and forest/grass areas are 13.6%, 9.81% and 62.5%, respectively. The main experiment site (100°03′58″ E, 26°02′43″ N, 1934 m a.s.l.) is located in Eryuan county on the northern side of Erhai Lake, Yunnan

province, China (Figure 1). A total of 60% of the water in Erhai Lake comes from inflows at the northern tip where the paddy-upland rotation fields are distributed on more than $224 \times 10^4$ ha, contributing approximately 26.6% of cropland. The study region covered more than 100 ha with a typical paddy-upland rotation. The Yong'an River crosses the downstream of this region, which is one of the primary three rivers flowing to Erhai Lake. The surface runoff water and responding rainfall in paddy-upland rotation were collected in Dali city (100°07′10″ E, 25°53′44.7″ N, 1968 m a.s.l.) to the west of Erhai Lake. In the southwest monsoon climate zone, the study areas were on a typical low latitude plateau. The annual mean temperature is 15.3 °C, and the mean annual precipitation is 1048 mm. Approximately 85%–96% of the precipitation is distributed in the rainy season from May to October [44].

The dominant source of nitrogen is agriculture in the catchment [45]. Rice seedlings were transplanted between late May and early June in southwest China, and the transplanting date ranged from 15 May to 22 May in 2017. Most fields were transplanted immediately followed by basal fertilization, with fewer fields fertilized approximately seven days after rice transplanting as top dressing or not fertilized. Urea or compound fertilizer mixed with urea (2:1) (60–200 kg N ha$^{-1}$) as the basal or top dressing fertilizer was used to fertilize the paddy fields from 15 May to 25 May. The cattle manure was applied by 1.5 t ha$^{-1}$; urea (46% N) and calcium superphosphate (12% $P_2O_5$) were applied before rice transplanting, and potassium sulphate (50% $K_2O$) was applied in the jointing stage. The cattle manure had a water content of 75%, N of 1.9%, $P_2O_5$ of 1.22% and $K_2O$ of 1.22%. Irrigation water was from a wetland in the upstream that flowed through a channel to the village and then to the cropland. The concentrations of TN, $NO_3^-$-N and $NH_4^+$-N in the irrigation water were 14.0 mg L$^{-1}$, 12.0 mg L$^{-1}$ and 0.42 mg L$^{-1}$, respectively. The inflow and outflow of the field always occurred simultaneously, so continuous irrigation water flooded the fields, with the fields full and water overflowing into ditches continuously.

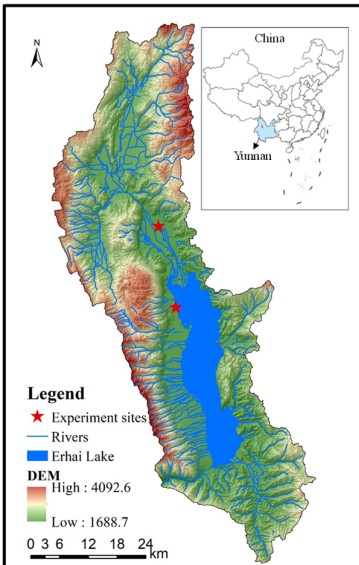

**Figure 1.** Locations of Erhai Lake Watershed and experimental site.

*2.2. Measurement of N Deposition*

Natural rainwater was collected manually immediately after precipitation events using a rain gauge cleaned by distilled water beforehand. The sealed lids of buckets were opened only when the rainfall happened. When the rainfall continued, daily samples (9:00 a.m. to 9:00 a.m. the next day) were collected and stored in clean polyethylene bottles. Some precipitation samples were discarded from the final analysis because of low vol-

umes (<10 mL) that did not permit a complete chemical analysis. The collected rain samples were frozen at −20 °C until analysis. The plastic buckets were cleaned with deionized water after each collection. The three sampling buckets were installed at the height of 1 m above the field surface in the center of this agricultural region. The concentrations of $NH_4^+$-N, $NO_3^-$-N and TN and rainfall depth were determined.

Simulated rainfalls were conducted on sunny days to obtain wet deposition and thus discuss the relationship between N deposition and $NH_3$ emission from soil after fertilization. Simulated rainfall events were performed from 8 May to 30 June in the center of the rice ecosystem region to mitigate the influences of transportation and human activities. A rainfall simulator (QYJY-501, Qingyuan Measurement Technology, Co. Ltd., Xi'an, China) was fixed in the paddy field to achieve a consistent rate of uniform rainfall. The rainfall intensities could be controlled by water pressures and nozzle sizes through the computer system. The rainfall nozzles were positioned 4 m aboveground and covered 25 $m^2$. Four groups of nozzles produced a rainfall with uniformity greater than 80%. The diameter of the simulated raindrops ranged from 0.37 mm to 6.0 mm, which was similar to natural raindrop in distribution and size [46].

Three collectors were placed under the covered region of the simulated system on a stand to avoid the interference of soil and rice plants but at a low height above the paddy water to flush $NH_3$ close to the ground. The simulated rainfall continued 20 min from 9:30 a.m. to 9:50 a.m. daily with uniform rainfall intensity except the natural rainfall days. When rainfall simulations were finished, water in collectors was collected into bottles to determine the concentrations of total N (TN), $NO_3^-$-N and $NH_4^+$-N.

The TN concentrations of water samples were determined before being filtered. Concentrations of TN were determined by alkaline potassium persulfate oxidation centrifugation and UV spectrophotometry [47]. Water samplings after being filtered through a 0.45 μm membrane were automatically pumped into an AA3 Autoanalyzer (Bran + Luebbe, Norderstedt, Germany) for determination of $NO_3^-$-N and $NH_4^+$-N concentrations by flow injection analysis technology [48].

Dry deposition of water-soluble gaseous or aerosol species and coarse and fine particles was collected using plastic buckets with a depth of 5 cm of distilled water as the surrogate surface [49]. Three plastic buckets were placed near the wet container at the height of 1 m above the field surface. The dry depositions were collected and flushed by distilled water into plastic bottles containing flush water at 9:00 a.m. every day. The samples were analyzed for $NH_4^+$-N, $NO_3^-$-N and TN concentrations. The dry deposition was calculated as

$$\text{Dry deposition (kg ha}^{-1} \text{ d}^{-1}) = C \times V/A \times 10^{-5} \tag{1}$$

where *C* is the concentration of nitrogen (mg $L^{-1}$), *V* is the volume of water (L) and *A* is the area covered by the container ($m^2$).

### 2.3. Measurement of NH₃ Volatilization

The $NH_3$ volatilization flux was measured with a continuous airflow enclosure method using a chamber [44]. The $NH_3$ volatilization collection device consisted of a chamber, a vent pipe, a chemical trap bottle and a vacuum pump, which were linked by plastic pipes to form a limited and confined space. A chemical trap bottle filled with 60 mL of 0.05 mol $L^{-1}$ $H_2SO_4$ was used to absorb the ammonia gas [50]. The air was pumped for 2 h in the morning (from 10:00 a.m. to 12:00 p.m.) and pushed the airflow through the $H_2SO_4$ for each treatment. The $NH_4^+$-N concentrations were determined for the ammonia absorbents by AA3 Autoanalyzer for $NH_4^+$-N determination [48]. We measured $NH_3$ volatilization from two fields fertilized by urea (46% N, 69 kg N $ha^{-1}$) with compound fertilizer (10% N, 45 kg N $ha^{-1}$) and compound fertilizer (15% N, 75 kg N $ha^{-1}$). There were three replicates per treatment. The daily ammonia volatilization flux was calculated from the average of the fluxes measured on each day. The daily measurement was conducted

every day from 10 May to 30 May, and then measurements were conducted every 2 days. The ammonia volatilization flux F was calculated as follows:

$$F = C \times V \times 14 \times 10^{-2} \times 24/(3.14 \times r^2 \times t) \tag{2}$$

where $F$ (kg ha$^{-1}$ d$^{-1}$) is the ammonia volatilization flux, $C$ (mol L$^{-1}$) is the NH$_4^+$-N concentration of the absorbing liquid, $V$ (mL) is the volume of dilute sulfuric acid as absorption liquid, $r$ (cm) is the semi-diameter of the chamber and $t$ (h) is the sampling time.

### 2.4. Surface Water Monitoring

The paddy field water and ditch water were sampled evenly within the rice growing region. The irrigation water was sampled from channels before they entered the fields during the irrigation period. A sample of paddy field water was composed of five sub-samples randomly collected from a field to obtain a composite sample at one sampling point in the study region. The water was sampled daily from 15 May to 31 May, then sampled every two days and later every five days. The runoff water drained from each plot using the runoff pool constructed near the plots in Dali city. The runoff samples were blended and taken after rainfall events. The water was stored in 200 mL polyethylene bottles for analysis of TN, NH$_4^+$-N and NO$_3^-$-N concentrations.

### 2.5. Statistical Analyses

The statistical analyses were performed, and graphs were prepared using the SPSS 19.0 statistical software package and Origin 9.0 software. Most of the data are presented as the mean ± SD (standard deviation) in the figures. Regression analyses were performed to evaluate the relationships between N deposition and NH$_3$ emissions. The significance of differences between the two periods was tested by the Kruskal–Wallis test at 95% confidence intervals. Differences were considered statistically significant at $p < 0.05$.

## 3. Results

### 3.1. NH$_3$ Volatilization

The temporal variations indicated that the NH$_3$ volatilization rate increased and reached the peak value two to four days after fertilization, and then a rapid decrease was observed at seven days after fertilization in the paddy field (Figure 2). The trends of NH$_3$ volatilization from the field fertilized by urea with a compound fertilizer exhibited similar patterns to those in the field fertilized by compound fertilizer, but the peak time appeared earlier. The cumulative NH$_3$ emissions from the fields fertilized by urea mixed with compound fertilizer and compound fertilizer were 16.1 kg N hm$^{-2}$ and 17.2 kg N hm$^{-2}$, accounting for 14.2% and 22.9% of N applied, respectively.

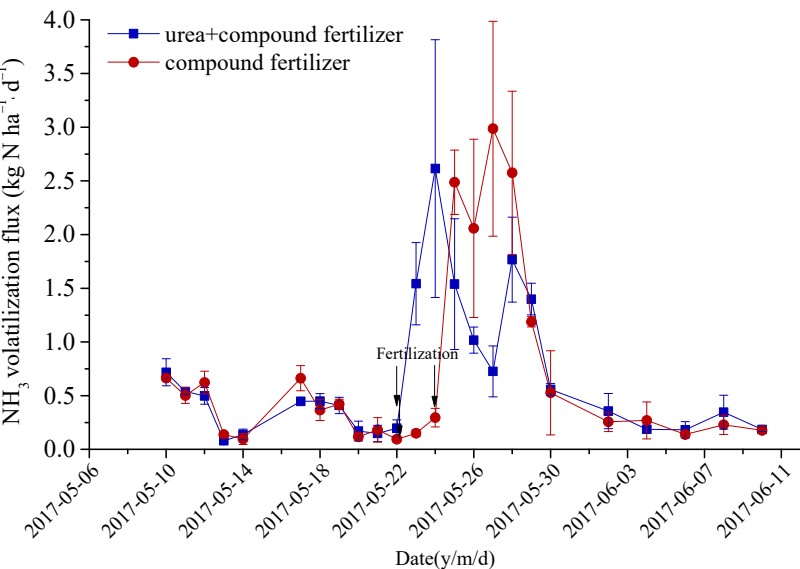

**Figure 2.** NH₃ volatilization from two fertilized paddy fields, whereas the fertilization dates were both on 22 May.

### 3.2. Wet and Dry Deposition of N

The concentrations of TN and $NH_4^+$-N in wet deposition showed large variations with the ranges of 0.14 mg L⁻¹ to 3.77 mg L⁻¹ and 0.10 to 1.50 mg L⁻¹, respectively, while the variation in $NO_3^-$-N concentrations was relatively stable (Figure 3). The fertilization period continued from 15 May to 25 May due to dispersed management. The high value of the volume concentration of $NH_4^+$-N appeared (averaged 1.07 mg L⁻¹) when the fertilization occurred and lasted six days after the completion of fertilizer application (approximately two weeks), which was increased by 153% compared with the average concentration in rainfall occurring in the later period of the rice growing season. The $NH_4^+$-N and $NO_3^-$-N concentrations were negatively related to rainfall depth ($p < 0.01$). Because of the high depth and frequent rainfall from July to September, the wet deposition of TN, $NH_4^+$-N and $NO_3^-$-N was 4.50 kg ha⁻¹, 1.37 kg ha⁻¹ and 0.55 kg ha⁻¹, respectively, which were higher amounts than those in May and June.

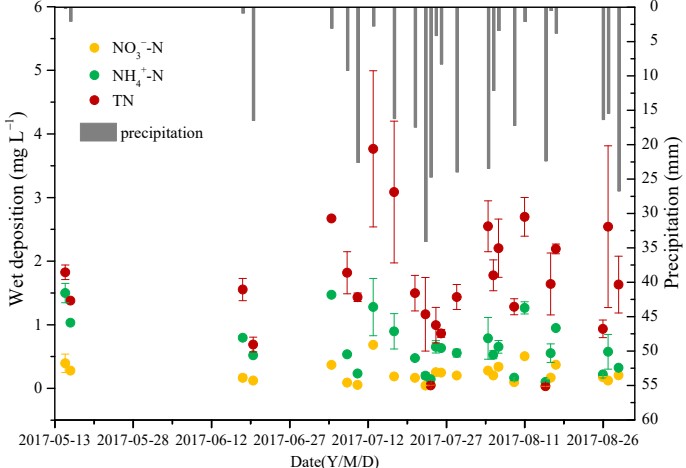

**Figure 3.** The temporal variations of N concentration in wet deposition and the corresponding natural rainfall depth.

The dry deposition rate of TN, NH₄⁺-N and NO₃⁻-N was 0.004–0.15 kg ha⁻¹ d⁻¹, 0.007–0.14 kg ha⁻¹ d⁻¹ and 0.002–0.02 kg ha⁻¹ d⁻¹, respectively (Figure 4a). The primary form was NH₄⁺-N, contributing more than 57% of TN. TN and NH₄⁺-N depositions increased substantially when the basal fertilizer application occurred, remained one week after the completion of fertilizer application and then declined, while NO₃⁻-N depositions were relatively stable during the experimental period. Moreover, the depositions of different N forms were all significantly positively related to NH₃ volatilization from cropland. Concentrations of NH₄⁺-N in simulated rainfall varied from 0.01 mg L⁻¹ to 0.79 mg L⁻¹, and the highest concentration occurred three to four days after the basal fertilization (Figure 4b). The NH₄⁺-N concentrations in simulated rainfall began to decline and remained stable one week after the completion of fertilization.

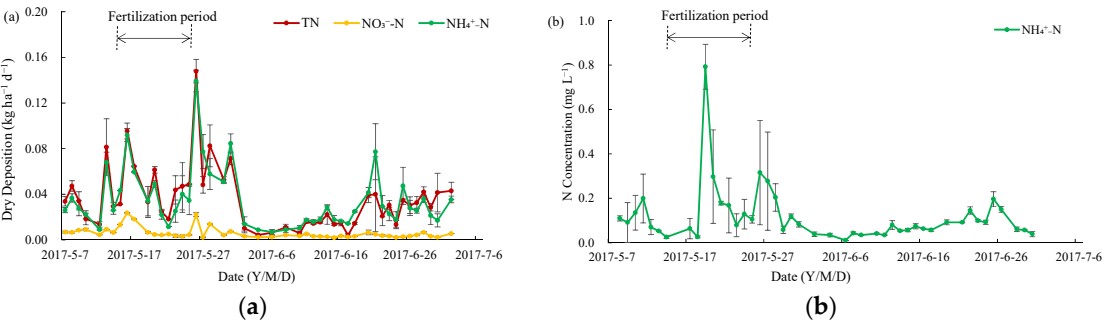

**Figure 4.** The temporal variations of N deposition in different forms and the associated responses to NH₃ volatilization in (**a**) dry deposition; (**b**) temporal variations of N concentration in simulated rainfall.

In the western part of Erhai Lake, the concentrations of TN, NO₃⁻-N and NH₄⁺-N in wet deposition showed large variations with the ranges of 0.77–1.84 mg L⁻¹, 0.18–0.48 mg L⁻¹ and 0.30–0.96 mg L⁻¹, respectively (Figure 5). The high value of the volume concentration of NH₄⁺-N appeared (averaged 0.96 mg L⁻¹ and 0.86 mg L⁻¹ in May and July, respectively) when the fertilization occurred, which was increased by 113% and 150% compared with the concentration in next month.

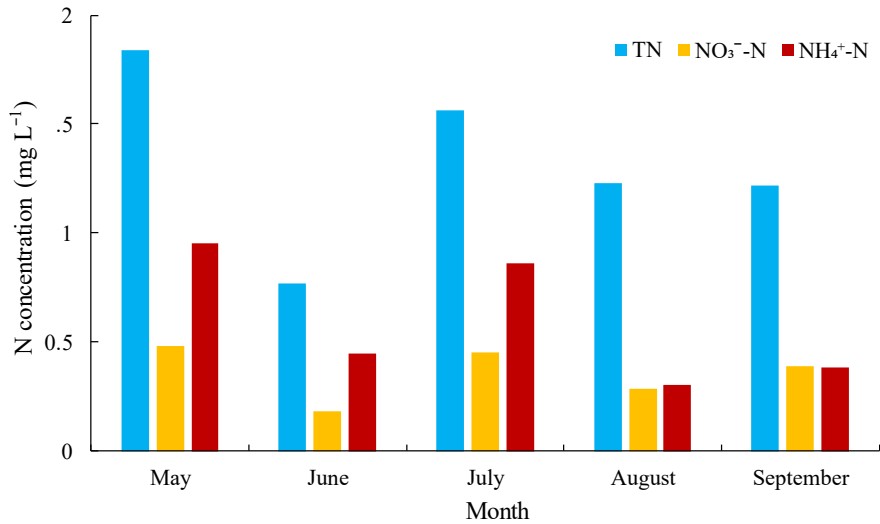

**Figure 5.** N concentration in precipitation wet deposition in different month in Dali city.

### 3.3. N Concentrations in Surface Water

The paddy field water was drained into ditches and then entered a tributary flowing to the lake. Concentrations of TN, $NO_3^-$-N and $NH_4^+$-N in paddy field water were in the ranges of 1.29–23.6 mg L$^{-1}$, 0.26–10.82 mg L$^{-1}$ and 0.06–5.51 mg L$^{-1}$, respectively (Figure 6a). Concentrations of TN, $NO_3^-$-N and $NH_4^+$-N were also substantially higher when the basal fertilizer application occurred than concentrations after this period and lasted two weeks after the completion of fertilizer application. The concentrations of TN, $NO_3^-$-N and $NH_4^+$-N in ditch water were significantly affected by paddy field water ($p < 0.01$). Concentrations of TN, $NO_3^-$-N and $NH_4^+$-N in ditch water varied from 2.84 to 22.6 mg L$^{-1}$, 1.71 to 19.5 mg L$^{-1}$ and 0.07 to 9.07 mg L$^{-1}$, respectively (Figure 6b). TN and $NO_3^-$-N concentrations in ditch water increased, while $NH_4^+$-N concentrations decreased compared with those in paddy field water. The primary N form in paddy field water was $NH_4^+$-N during fertilization period until one week after the completion of fertilizer application, and $NO_3^-$-N in paddy field water and ditch water contributed 17.8%–95.9% of TN in the entire experimental period.

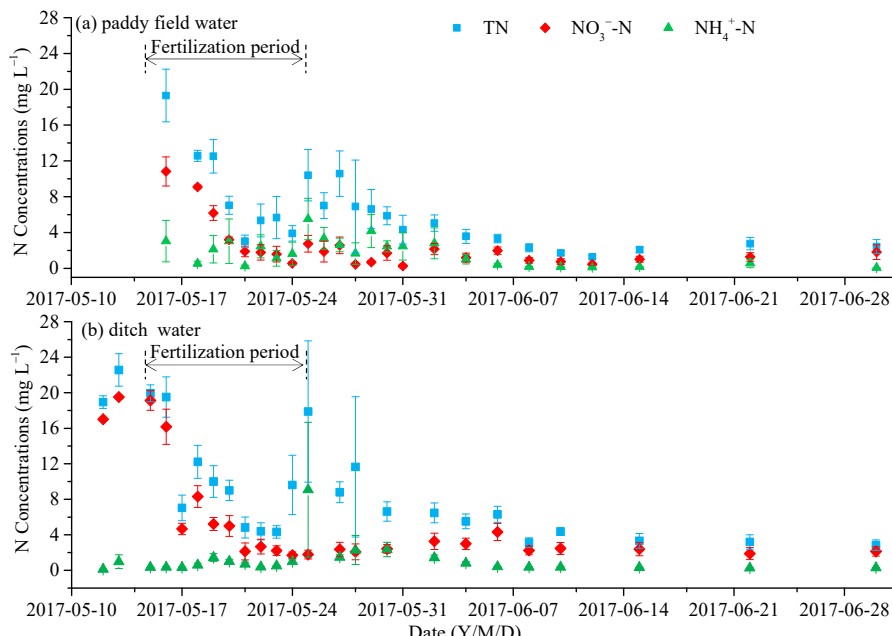

**Figure 6.** TN, $NO_3^-$-N and $NH_4^+$-N concentrations in: (**a**) paddy field water; (**b**) ditch water.

The concentrations of TN, $NO_3^-$-N and $NH_4^+$-N in runoff water were in the ranges of 2.55–4.78 mg L$^{-1}$, 1.06–3.26 mg L$^{-1}$ and 0.83–0.20 mg L$^{-1}$, respectively (Figure 7). Concentrations of TN, $NO_3^-$-N and $NH_4^+$-N were higher in May when the N fertilizer application occurred than concentrations from June to September, respectively.

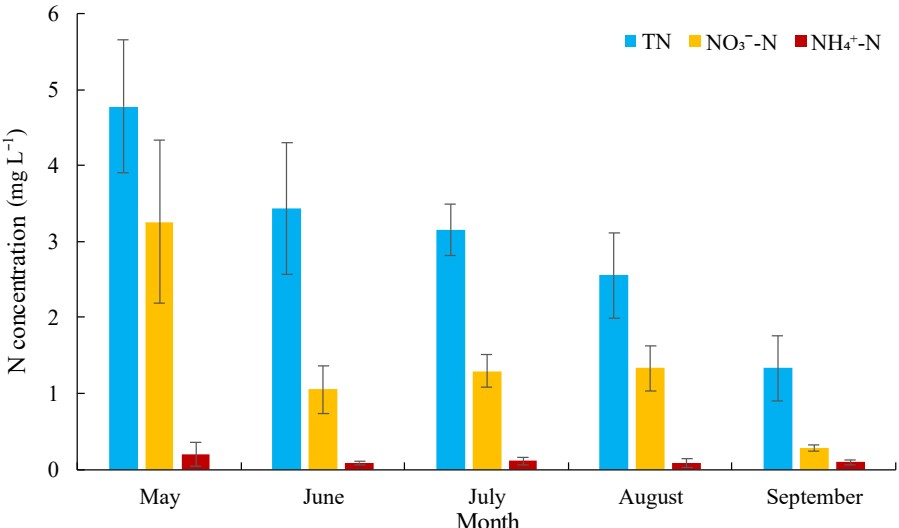

**Figure 7.** TN, $NO_3^-$-N and $NH_4^+$-N concentrations in runoff water.

The flow at the ditch outlet was the primary source of N input to the tributary flowing to the lake, influenced directly by ditch water ($p < 0.01$). TN and $NH_4^+$-N concentrations of outlet water ranged from 2.63–20.1 mg $L^{-1}$ (average 7.71 mg $L^{-1}$) to 0.50–3.06 mg $L^{-1}$ (average 1.37 mg $L^{-1}$) during fertilization period until two weeks after the completion of fertilizer application, respectively, and were higher by 377% and 322% than concentrations after this period, respectively ($p < 0.05$) (Figure 8). Accordingly, concentrations of TN, $NO_3^-$-N and $NH_4^+$-N from ditch water to outlet water decreased by 14.5%, 41.3% and 13.6%, respectively.

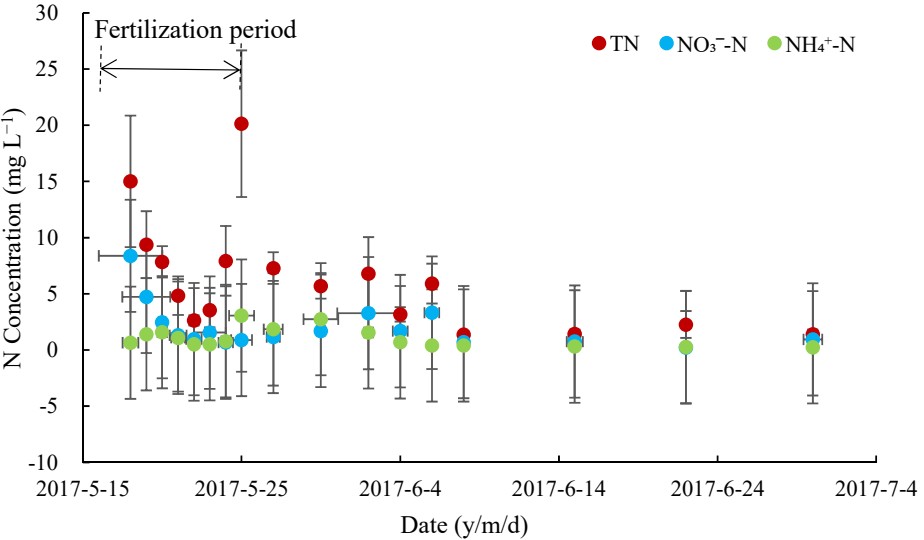

**Figure 8.** TN, $NO_3^-$-N and $NH_4^+$-N concentrations in outlet of ditch water.

### 3.4. Critical Period of Nitrogen Load via Deposition and Runoff

The critical period of higher concentrations or depositions of N (CP period) was during the fertilization period up until one to two weeks after the completion of fertilizer application compared with the other periods (general period, defined as the GP period). Based on the division of periods, the average volume-weighted concentrations of $NO_3^-$-N and $NH_4^+$-N in the CP period were 0.29 mg $L^{-1}$ and 1.07 mg $L^{-1}$, respectively, which were

95.9% and 114% higher, respectively, than those in wet deposition in the GP period ($p <$ 0.01) (Figure 9a). However, the TN concentration was not different between the two periods. The mean $NH_4^+$-N concentration in deposition from cropland accounted for 54.5% of atmospheric $NH_4^+$-N deposition in natural rainfall. For dry deposition, the mean depositions of $NO_3^-$-N, $NH_4^+$-N and TN in the CP period were 0.009 kg ha$^{-1}$ d$^{-1}$, 0.055 kg ha$^{-1}$ d$^{-1}$ and 0.058 kg ha$^{-1}$ d$^{-1}$, respectively, which were 144%, 120% and 155% higher, respectively, than those in the GP period ($p < 0.01$) (Figure 9b). The cropland contributed approximately 53.4% of dry deposition of $NH_4^+$-N. The higher period of TN and $NH_4^+$-N concentrations in outlet water was prolonged one week compared with that for deposition. The average $NO_3^-$-N concentration in outlet water in the CP period was 52.8% higher than that in the GP period (Figure 9c).

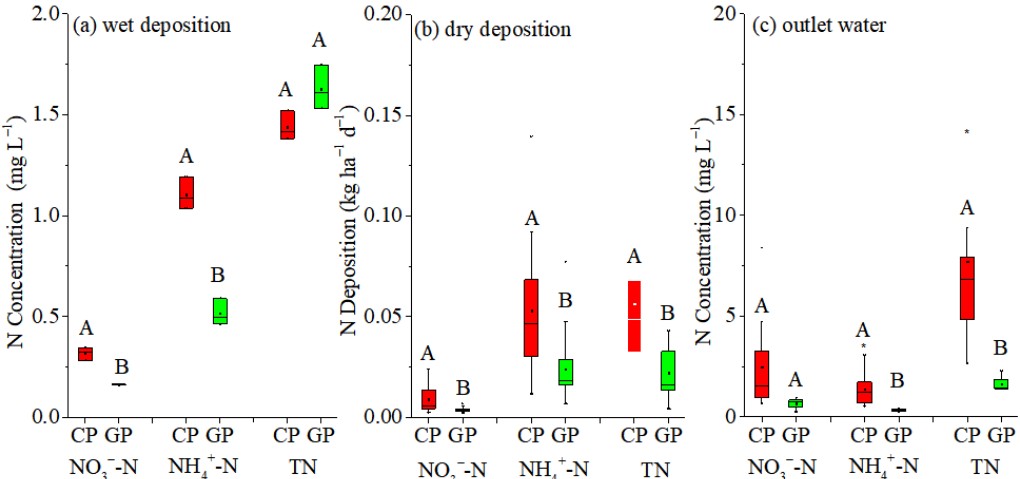

**Figure 9.** N concentration of: (**a**) wet deposition; (**b**) dry deposition; (**c**) outlet water, in two periods. Bars with different lowercase letters and capital letters indicate substantial differences between CP and GP at $p < 0.05$. CP: the critical period with high deposition or concentration; GP: the general period compared with CP. In the box-plots, the small square lattice represents the arithmetic mean of the data. Boxplots indicate the 25th, 50th, and 75th percentiles and whiskers indicate the minimum and maximum values.

### 3.5. Relationship between N Deposition and NH₃ Volatilization

The dry depositions of TN, $NH_4^+$-N and $NO_3^-$-N in the rice ecosystem were positively related to $NH_3$ volatilization rates from cropland (Figure 10a) ($p < 0.05$). The wet depositions of TN and $NH_4^+$-N were positively related to $NH_3$ volatilization rates, but only TN depositions had a significant relationship to $NH_3$ volatilization rates (Figure 10b) ($p < 0.05$).

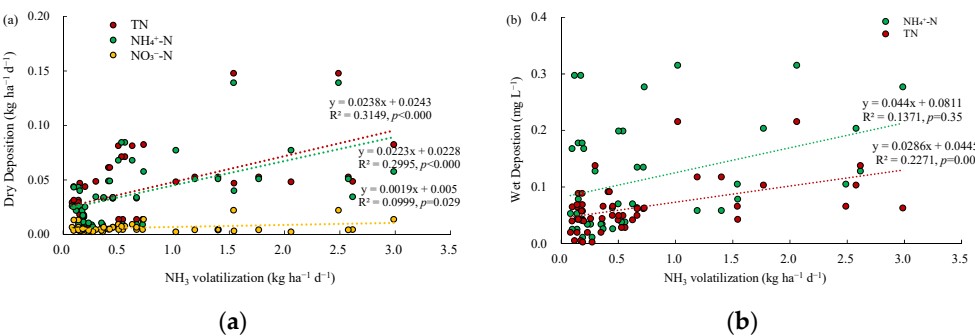

**Figure 10.** The relationships between $NH_3$ volatilization and (**a**) N dry deposition and (**b**) wet deposition.

## 4. Discussion

### 4.1. Relationship between NH₃ Volatilization and N Deposition

Positive correlations were found between the depositions of TN and $NH_4^+$-N and $NH_3$ volatilization from cropland (Figure 10a,10b). These correlations are consistent with those of a previous study across all land use types in China conducted by Xu et al. [51], who reported that positive correlations were also found between $NH_3$ emissions and $NH_x$ deposition fluxes across 43 monitoring sites. It has been reported that about 90% of all atmospheric $NH_3$ is from a local source, and the ambient $NH_3$ concentration is the most useful parameter for evaluating changes in $NH_3$ emission [52]. Agriculture activities are the largest sources of $NH_3$ emissions because of animal husbandry and $NH_3$-based fertilizer applications [53,54]. Previous studies showed that sites within major agricultural regions have experienced larger increases in $NH_4^+$ concentrations than those of urban-affected sites [55,56]. Therefore, high-intensity $NH_3$ emissions in farmland regions lead to a higher concentration of $NH_3$ in the atmosphere than in other regions [57–60]. For example, $NH_3$ concentrations increased sharply after N fertilizer application related to $NH_3$ volatilization in both the North China Plain and a double rice region in subtropical China [21,22,61,62]. Because of the short atmospheric lifetime and high dry deposition velocity, $NH_3$ typically deposited near the emission source, and $NH_3$ concentrations generally decay exponentially some distance away from a source because of dispersion and dilution [63]. For example, Xu et al. [64] showed that the concentration of $NH_3$ decreased by 64% at a 640 m distance from its source, while Asman et al. [65] found a 70% reduction at a 4 km distance from the source. In agroecosystems, variations in monthly $NH_4^+$-N concentrations are much greater than those for $NO_3^-$-N, primarily related to $NH_3$ and $NO_x$ emissions and precipitation [14].

### 4.2. Critical Period of Nitrogen Deposition and Runoff

The paddy fields affected both wet and dry deposition by $NH_4^+$-N primarily (Figures 3–5). This result is consistent with the findings by Cui et al. [14] who showed that $NH_4^+$-N contributed 68% of wet inorganic N deposition in southeast China. $NH_4^+$-N originated from agricultural emissions using $\delta^{15}$N-$NH_x$ values [66]. In this study, high $NH_4^+$-N depositions were observed in the fertilization period and continued one week after the completion of fertilizer application, while low and stable N depositions occurred after this period (Figures 3 and 4). The $NH_3$ volatilization rate reached a peak one to three days after fertilization and then decreased to a low level from the peak value after seven to ten days in a paddy field [20]. In this study, the fertilization of paddy fields primarily occurred in the first ten days after basal fertilization, and $NH_3$ emissions also declined one week after fertilization (Figure 2) and decreased with the decline of $NH_4^+$-N concentrations in field water ($p < 0.01$). Temporal variations in $NH_3$ emissions due to fertilization led to large temporal variations in $NH_4^+$-N depositions. In addition, the duration of fertilization in the ecosystems determined the length of the critical period. Moreover, differences in the N deposition between the two periods were particularly large (Figure 9). Liu et al. [67] showed that N deposition rates in April, June, July and August when farmers apply N fertilizer are often greater than those in other months. N fertilizer use is significantly related to wet N deposition [27,28]. Therefore, the critical period for N deposition was the fertilization period, which continued one week after the completion of fertilizer application in the rice ecosystem.

N concentrations in both paddy field water and ditch water increased in the fertilization period and continued two weeks after the completion of the fertilizer application (Figures 6 and 8). During the rice growing season, runoff water is generated when the volume of rainfall combined with the field ponding water exceed the capacity of field berms to enclose water [68]. The duration between applications of different fertilizer sources and a runoff event can affect the concentrations of nutrients in runoff [69,70]. A

portion of the N from watersheds entering river networks, particularly after applying fertilizers, is nutrients that are easily lost to channels. That was consistent with the observations of runoff water (Figure 7). A previous study also showed that TN loss from agricultural land via runoff increased with increasing N fertilizer application rates [41]. However, Tang et al. [71] found that N concentration in the overflow water was very high within approximately a week after fertilization in a paddy field. However, people with less land and decentralized operation in the Erhai Lake Watershed tended to transplant and fertilize rice one after another for approximately ten days, which prolonged the influencing time of agricultural activities on the water quality. Moreover, the attenuation coefficients of the ditch in the paddy ecosystem for TN, $NO_3^-$-N and $NH_4^+$-N were low due to the short distance (Figure 8). Therefore, the measures in the tributaries and their outlets were needed to reduce the effect of cropland on the surrounding water body and lake.

Precipitation and runoff are the two primary pathways of water supply for Erhai Lake, and so ditches and deposition are the main pathways of N input derived from cropland [72]. A previous study reported that approximately more than 65% of the agricultural nonpoint source N enters the ditch system distributed extensively in a catchment dominated by paddy fields [73]. In addition, the N concentrations of irrigation water were distinctly lower than N concentrations of the outlet of ditch water (Figure 9). Hua et al. [74] also reported on the purification role of rice fields. For the Yunnan Plateau lakes, the paddy ecosystem has a certain purification effect on water with pollution from rural areas. However, from July to September when most of the heavy rainstorms occurred, rainfall runoff was the primary driving force and carrier of the N output and high field N runoff levels, and N enrichment in the drainage water typically occurs synchronously [4,75,76]. Thus, the large amount of irrigation water and fertilizers in May and heavy rainfall from July to September were the primary factors influencing the N concentration of Erhai Lake.

Lakes in southwest China are in the initial stage of eutrophication and primarily affected by agricultural activities, which are different from those of lakes in floodplains (e.g., Taihu Lake and Chaohu Lake) in China [36,77]. Currently, in southwest China, 77.8% of the Yunnan Plateau lakes are eutrophic with high nutrient or low nutrient inputs [36]. Important drivers of lake eutrophication are agricultural runoff and nitrogen deposition [35,78,79]. In southwest China, temperature, light and trophic conditions of lakes from May to September can meet the requirements for the growth of algae or cyanobacteria [80]. For example, clear peaks of *T. bourrellyi* cell abundance occur in July and August every year, and monthly variations in different forms of N and P concentrations indicate that Erhai Lake can be in a eutrophic state from July to September [81]. Late May and early June is the starting point of the peaks of algae or cyanobacteria growth for lakes in southwest China, which was in accordance with the critical period of N output from cropland by deposition and runoff. Moreover, water that flows from mountains to lakes generally has a relatively fast velocity as the water passes through croplands, leading to nutrients losses. Therefore, controlling N input through runoff and deposition derived from the rice ecosystem was critical in fertilization period and continued one to two weeks after the completion of fertilizer application.

### 4.3. Suggestions for Agricultural Measures

Based on this study, the recommendation is that the water that flows from paddy fields into drainage ditches should be intercepted/contained while maintaining a paddy field water at a depth of 5 cm to 15 cm. For example, a paddy field can be partially blocked by increasing field berms. Water retention time will increase, and that is beneficial for nutrient reduction [82,83]. Additionally, ecological ditches and improved zeolite-ecological ditches can be used to intercept nutrient-enriched flow pathways between agricultural fields and streams in paddy ecosystems [84]. In this study, N deposition in agricultural regions is strongly dependent on the temporal pattern of $NH_3$ emission within the region (Figure 10a,b). The ways to control N deposition are related to the sources of $NH_3$ emis-

sions. Proper N fertilizer management, including optimal and deep fertilization, is effective to decrease $NH_3$ emission [85] and prevent N deposition from entering into the surrounding water. The large-scale farms may reduce the duration of fertilization because of modern agricultural equipment and thus reduce the time of high N losses [86].

## 5. Conclusions

In this study, the dynamic N concentration of surface runoff and N deposition derived from rice ecosystem were found frequently in a watershed within a lake system in southwest China. Our findings indicate that for the rice ecosystem, the critical period of N outputs via deposition and runoff derived from cropland was in the fertilization period and continued one to two weeks after the completion of the fertilizer application. Moreover, cropland contributed more than 54% of $NH_4^+$-N in deposition in the critical period, being positively related to $NH_3$ emission due to fertilization. Therefore, a decrease in agricultural $NH_3$ volatilization and interception of runoff from paddy fields into drainage ditches in the critical period are needed. The conducted work is beneficial for the overall water quality and conservation practices of lakes and reservoirs in rice growing areas.

**Author Contributions:** Conceptualization, S.G., L.Z., H.Y. and H.L.; Methodology, S.G. and W.L.; Software, S.G. and T.Y.; Validation, S.G. and L.Z.; Formal analysis, S.G.; Investigation, S.G.; Resources, L.Z. and H.L.; Data Curation, S.G. and T.Y.; Writing—original draft preparation, S.G. and T.Y.; Writing—review and editing, H.Y. and J.L.; Supervision, L.Z. and H.L.; Project Administration, S.G.; Funding acquisition, L.Z. and S.G. All authors have read and agreed to the published version of the manuscript.

**Funding:** This work was financially supported by the National Natural Science Foundation of China (42067047).

**Data Availability Statement:** The datasets used and/or analyzed during the current study are available from the corresponding author on reasonable request.

**Conflicts of Interest:** The authors declare no conflict of interests.

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
