# Peer review of "Nitrogen Transport/Deposition from Paddy Ecosystem and Potential Pollution Risk Period in Southwest China"

_water, doi:10.3390/w14040539_

Round 1

Reviewer 1 Report

Abstract:

(1) Deposition is not a pathway of nitrogen losses from cropland.

(2) The author concluded that “Therefore, decrease NH3 emissions, agriculture driven N depositions and intercept water flow from paddy fields into drainage ditches in the critical period can significantly impact the agricutural nutrient losses and lake eutrophication.” However, there was no evidence that NH3 emissions, agriculture driven N depositions impact the lake eutrophication. Therefore, I suggested that the conclusion need be summarized according with the topic of this study.

(3) “NH3 emissions” is not correct. Maybe NH3 volatilization?

Introduction:

(1) There is no topic in the part of Instroduction. Therefore, the structure of Introduction is chaos.

(2) The review about the research progress in the world, especially in Erhai lake, is not enough.

(3) The key scientific question is not clear and need to be further summarized.

Materials and methods:

(1) Figure 1 is not standard. Especially the Chinese map.

(2) the mothod of the measure of wet and dry deposition should be merged.

(3) What is the concentration of H2SO4? Why not boric acid?

(4) Is there replication of the two treatments?

(5) Is there corralation of surface water monitoring and NH3 volatilization?

(6) How to asssess the relationship between N deposition and NH3 emission from soil after fertiliza by simulated rainfalls? There was not any information about the water source of simulated rainfalls.

Results and Disscusion:

(1) Is there difference between Figure 6 and 8 about the concentrations in ditch water? And there is not standard deviation in figure 8.

(2) the section of “Relationship between N depostion and NH3 volatilization” should be transport into the secetion of “Discussion”.

(3) How to understand the correlation of N deposition, NH3 volatilization, and surface water quality?

(4) The topic of Disscussion is not clear. It is very diffcult to understand the content of this section. Disscussion should be the enhancement of results.

Conclusions:

(1) Conclusions need to be drawn from the research in this paper. Some conclusions maybe Inappropriate. For example, the author think that “optimal fertilization and paddy water management are 417 required to decrease NH3 emissions and intercept water flow from paddy fields into 418 drainage ditches.” However, there is not optimal fertilization and paddy water management in this study.

Reviewer 2 Report

See the attached PDF
